# Steatosis and Metabolic Disorders Associated with Synergistic Activation of the CAR/RXR Heterodimer by Pesticides

**DOI:** 10.3390/cells12081201

**Published:** 2023-04-21

**Authors:** Yannick Dauwe, Lucile Mary, Fabiana Oliviero, Marina Grimaldi, Patrick Balaguer, Véronique Gayrard, Laïla Mselli-Lakhal

**Affiliations:** 1Toxalim (Research Centre in Food Toxicology), Université de Toulouse, INRAE, ENVT, INP-Purpan, UPS, 31027 Toulouse, France; 2Institut de Recherche en Cancérologie de Montpellier, Inserm U1194-Université Montpellier-Institut régional du Cancer Montpellier, CEDEX 5, F-34298 Montpellier, France

**Keywords:** CAR, RXR, nuclear, receptor, pesticides, steatosis, NAFLD, tributyltin, metabolism, synergism

## Abstract

The nuclear receptor, constitutive androstane receptor (CAR), which forms a heterodimer with the retinoid X receptor (RXR), was initially reported as a transcription factor that regulates hepatic genes involved in detoxication and energy metabolism. Different studies have shown that CAR activation results in metabolic disorders, including non-alcoholic fatty liver disease, by activating lipogenesis in the liver. Our objective was to determine whether synergistic activations of the CAR/RXR heterodimer could occur in vivo as described in vitro by other authors, and to assess the metabolic consequences. For this purpose, six pesticides, ligands of CAR, were selected, and Tri-butyl-tin (TBT) was used as an RXR agonist. In mice, CAR’s synergic activation was induced by dieldrin associated with TBT, and combined effects were induced by propiconazole, bifenox, boscalid, and bupirimate. Moreover, a steatosis, characterized by increased triglycerides, was observed when TBT was combined with dieldrin, propiconazole, bifenox, boscalid, and bupirimate. Metabolic disruption appeared in the form of increased cholesterol and lowered free fatty acid plasma levels. An in-depth analysis revealed increased expression of genes involved in lipid synthesis and lipid import. These results contribute to the growing understanding of how environmental contaminants can influence nuclear receptor activity and associated health risks.

## 1. Introduction

Non-alcoholic fatty liver disease (NAFLD) and non-alcoholic steatohepatitis (NASH) are currently the main global causes of liver disease, with an estimated prevalence of 25.24% for NAFLD [1]. They can cause a range of liver anomalies, from a benign fat accumulation, known as steatosis, to inflammation with or without fibrosis, known as NASH, which can evolve into cirrhosis and hepatocellular carcinoma [2]. Two types of steatoses have been distinguished. Macrovesicular steatosis is characterized by a single large vacuole that pushes the nucleus to the membrane. In this case, cellular ballooning can also happen [3]. This type of steatosis is associated with an increase in de novo fatty acid synthesis, depleted lipid export, depleted gluconeogenesis, and a high-fat diet [4,5]. Microvesicular steatosis is characterized by a multitude of small lipid vesicles that do not displace the nucleus. This type of steatosis has been linked to the dysfunction of mitochondrial fatty acid peroxidation [6].

NAFLD prevalence has doubled in the last two decades, reaching 30% in developed countries [7]. Among the affected population, 50% are obese, 69% have dyslipidemia, and 22.5% have diabetes. Moreover, 40% of NAFLD’s evolve into NASH, with a mortality rate of 0.77/1000 [1]. NAFLD’s associated etiologies are metabolic syndrome and obesity [8]. However, obesity alone cannot explain all NAFLD cases. During the diagnosis of steatohepatitis, once alcoholism and metabolic syndrome have been excluded, secondary causes must be explored. Various factors can contribute to NAFLD, such as infections (hepatitis C virus), endocrine disorders (hypothyroidism and hypopituitarism), coeliac disease, genetic diseases (Wilson disease and A1AT deficiency), drugs (methotrexate, tamoxifen, corticosteroids, etc.), and environmental contaminants [9].

Environmental contaminants are suspected to play a part in this pathology or in the evolution from NAFLD to NASH. Recently, the acronym TAFLD (toxic associated fatty liver disease) has been used in the literature. TAFLD suggests that environmental contaminants can play a part in the development of steatosis [10]. In fact, experimental and epidemiological data show that numerous chemicals, especially endocrine disruptors, play a part in the development and evolution of NAFLD [10,11]. Pesticides represent a significant share of steatogenic compounds [12,13,14].

The implications of nuclear receptors in TAFLD have been demonstrated in various studies [15,16]. Indeed, the activation of hepatic nuclear receptors by pollutants is known to be associated with mechanisms that contribute to steatosis, such as increased de novo lipogenesis, decreased fatty acid oxidation, increased hepatic lipid uptake, and decreased gluconeogenesis [9,17]. As part of this work, we investigated the xenosensor, constitutive androstane receptor (CAR), and its heterodimerization partner, retinoid X receptor (RXR). This nuclear receptor is expressed in the liver and is particularly interesting to study since it regulates detoxification, endocrine function, and energy metabolism genes, and is therefore at the crossroads between detoxification and energy metabolism [18,19,20]. CAR is known to modulate the expression of key lipogenic genes such as fatty acid synthase (*Fasn*), and of lipid catabolism-limiting enzymes such as carnitine palmitoyl transferase 1A (*Cpt1a*) [21]. The activation of this receptor leads to lipid accumulation in the liver [22,23]. For these reasons, CAR is considered a mediator of metabolic disruption induced by environmental contaminants [21,24].

One of the main concerns in toxicology is the risk assessment of cocktails of environmental contaminants, as they can have additive or even synergistic effects. Recent studies have shown that different chemicals can bind cooperatively to the ligand-binding pocket of Pregnane X Receptor (PXR), leading to synergic activation of the receptor by a mixture of chemicals that would have very low efficacy if administered separately [25,26]. These studies also suggest that the environmental ligands of PXR and RXR can act together to induce cooperative recruitment of the steroid receptor coactivator-1 (SRC-1) by the heterodimer, leading to the synergic activation of PXR–RXR target gene expression [25].

The main objective of this study is to investigate whether the synergistic activation of nuclear receptors by environmental contaminants can occur in vivo, and to assess the consequences on the development of metabolic disorders. To achieve this, this study assesses the co-activation of the CAR-RXR heterodimer by pesticides that activate CAR in combination with Tri-butyl-tin (TBT), a known RXR agonist [27,28,29,30]. The activation of the heterodimer is measured by analyzing the expression of *Cyp2b10,* which is a prototypical target gene of CAR. The consequences of this activation on metabolic disruptions are assessed by measuring the expression of genes involved in lipid metabolism and in lipid accumulation in the liver.

## 2. Materials and Methods

### 2.1. Cell Culture

For cell culture related experiments, chemicals, culture medium and serum were purchased from Invitrogen (Waltham, MA, USA). To characterize the specificity of chemicals in mouse CAR (mCAR), we established an mCARreporter cell line. HG5LN are HeLa cells that were stably transfected by a (GAL4RE)5-bGlob-Luc-SVNeo plasmid [31]. These cells were stably transfected by a pSG5-GAL4 DNA Binding Domain–mCAR Ligand Binding Domain–puromycin plasmid, and in the presence of 0.5 μg/mL puromycin. Three weeks after the initiation of puromycin selection, the maximal luciferase expression of resistant clones was measured in the presence of 10 μM TCPOBOP and 0.3 μM luciferin for 24 h. Two days later, the minimal luminescence expression from the same individual clones was measured without the ligand. In total, 5 to 10 clones were chosen for their ligand inducibility of luciferase expression. The most inducible clones were expanded and rechecked for inducibility, and aliquots were frozen at different stages. One clone per receptor was maintained in culture and used for ligand screening. HG5LN cells were cultured in Dulbecco’s modified Eagle’s medium (DMEM): Nutrient Mixture F-12 (DMEM/F-12) containing phenol red and 1 g/L glucose; then, they were supplemented with 5% fetal bovine serum, 100 units/mL of penicillin, 100 μg/mL of streptomycin, and 1 mg/mL geneticin in a humidified 5% CO_2_ atmosphere at 37 °C. HG5LN mCAR cells were cultured in the same culture medium supplemented with 0.5 μg/mL of puromycin.

### 2.2. Luciferase Assay

HG5LN mCAR reporter cells were seeded at a density of 40,000 cells per well in 96-well white opaque tissue culture plates (CellStar®, Greiner Bio-One, Les Ulis, France) in Dulbecco’s Modified Eagle Medium: Nutrient Mixture F-12 (DMEM/F-12) with phenol red and 1 g/L glucose; then, they were supplemented with 5% stripped fetal bovine serum, 100 units/mL of penicillin, and 100 μg/mL of streptomycin (test medium). The chemicals to be tested were added 24 h later, and the cells were incubated at 37 °C for 16 h. The experiments were performed in quadruplicate. At the end of the incubation period, the culture medium was replaced with a test medium containing 0.3 mM luciferin solution. Luciferase activity was measured for 2 s in intact living cells after 10 min stabilization using a Micro Beta Wallac luminometer (PerkinElmer, Waltham, MA, USA). The EC50 values were calculated using GraphPad Prism 9 (GraphPad Software Inc, San Diego, CA, USA).

### 2.3. Animal Experiment

The in vivo study was conducted in accordance with the European Union guidelines for laboratory animal use and care. An independent ethics committee (Toxcométhique, INRAE ToxAlim, Toulouse, France) approved the experiment (Approval Code: Toxcom247). For this experiment, eight-week-old male C57BL/6 J mice from the Janvier Lab (Le Genest-Saint-Isle, France) were used. They were housed in polycarbonate cages Type S (Charles River, Ecully, France) at a temperature of 20 °C to 24 °C, under a 12-h light/dark cycle, with ad libitum access to food and water. The housing was enriched with a stainless steel hut to provide shelter and diminish stress. Fourteen groups of six mice were force-fed once a day for 3 days. The treatments used were dimethyl sulfoxide (DMSO), TBT (5 mg/kg/day), dieldrin (0.5 mg/kg/day ± TBT), propiconazole (50 mg/kg/day ± TBT), boscalid (50 mg/kg/day ± TBT), bifenox (50 mg/kg/day ± TBT), bupirimate (50 mg/kg/day ± TBT), and pendimethalin (50 mg/kg/day ± TBT). DMSO and all pesticides were purchased from Sigma-Aldrich (Saint-Quentin Fallavier, France).

After 3 days, each mouse’s body weight was measured. Then, a blood sample was taken from the submandibular vein using a lancet and placed in an EDTA-coated tube (BDMicrotainer^®^; BD, Le Pont-de-Claix, France). Plasma was obtained via centrifugation (1500 g for 10 min at 4 °C) and stored at −80 °C for biochemical analysis. Then, the animals were euthanized via cervical dislocation. The mice’s livers and subcutaneous and epidydimal white adipose tissues were collected, weighed, snap-frozen in liquid nitrogen and stored at −80 °C for further use.

### 2.4. Gene Expression Studies

RNA extraction was performed using Tri reagent (Molecular Research Center inc., Cincinnati, OH, USA) following the method of Chomczynski et al. [32]. A Nanodrop^®^ (Thermo Scientific, Waltham, MA, USA) was used to determine RNA concentration. The retrotranscription of 2 µg RNA was performed using an Applied Biosystems high-capacity cDNA reverse transcription kit (Thermo Fisher, Waltham, MA, USA). Quantitative Polymerase Chain Reaction (qPCR) was conducted using SYBR Green (Applied Biosystems^®^, Waltham, MA, USA) and the primers presented in Table 1.

The thermal cycler C1000Touch from Biorad (Marnes-La-Coquette, France) (CFX96) was used for qPCR and was programmed as follows: first, a cycle of 10 min at 95 °C; then 39 cycles of 15 s at 95 °C and 30 s at 60 °C; and then, a melt curve from 65 °C to 95 °C for 0.5 °C/5 s. The qPCR data were normalized using TATA-box-binding protein (*Tbp*) mRNA levels and analyzed using LinRegPCR (2015.3 version).

### 2.5. Liver Neutral Lipid Analysis

The hepatic neutral lipid contents were determined as previously described in [33]. The liver samples were homogenized in methanol/5 mM EGTA (2:1, *v*/*v*), and the lipids were extracted with chloroform/methanol/water (2.5:2.5:2.1, *v*/*v*/*v*) in the presence of glyceryl trinonadecanoate, stigmasterolm and cholesteryl heptadecanoate (Sigma) as internal standards. The triglycerides (TG), free cholesterol, and cholesterol esters were analyzed via gas–liquid chromatography using a Focus Thermo Electron system from Thermo Scientific (Pittsburgh, PA, USA).

### 2.6. Histology

The previously collected liver samples embedded in Tissutek OCT (Sakura Finetek, Alphen aan de Rijn, The Netherlands) were used for this part of the experiment. The slices were cut using a Leica CM1900 cryostat (Leica Biosystems, Nanterre, France). The blade temperature was fixed at −10 °C, and the enclosure was fixed at −17 °C. Three cuts per subject, 9 µm in thickness, were collected, placed on glass slides (Superfrost Thermo Scientific, Waltham, MA, USA), and dried for one hour. Then, the slides were fixed in a 4% formalin/phosphate buffer solution (PBS) (Sigma) for 45 min and rinsed in PBS for 20 s and in isopropanol 30% (Sigma) for 20 s. Next, the slides were placed in a 10-min bath of red oil stain (Sigma) and rinsed for 30 s with isopropanol 30%. A second stain was applied for 3.5 min using Harris Haematoxylin (Sigma), followed by a rinsing for 10 min with running water. Finally, the slides were assembled using Aquatex and dried for one week. The slides were scanned using a Nanozoomer digital pathology scanner (Hamamatsu, Shizuoka, Japan) and visualized using a Hamamatsu NDP viewer. ImageJ public domain software (ImageJ Website, https://imagej.net/ij/, accessed on 5 January 2023) was used to assess the area covered by lipid droplets. Lipid vesicle coverage equal to or greater than 5% was considered to represent steatosis. Histological scoring was performed for steatosis using a previously established scoring system [34].

### 2.7. Plasma Analysis

Alanine aminotransferase (ALAT), aspartate amino transferase (ASAT), TG, free fatty acids (FFA), cholesterol, high-density lipoprotein (HDL), and low-density lipoprotein (LDL) levels were established using an ABX Pentra 400 biochemical analyzer (Horiba Medical, Anexplo facility, Toulouse, France).

### 2.8. Determination of Combined Effects and Statistical Analysis

First, all the results were transformed into the fold change of the DMSO-treated group. Graphpad Prism 9 was used to carry out statistical analyses. Each dataset was analyzed using a one-way ANOVA test accompanied by a Tukey’s multiple comparisons test, with *p* < 0.05 considered significant. To identify the combined effects, we considered the “Combination Subthresholding approach” and the “highest single agent approach/cooperative effect”, as previously described in [35]. Briefly, the “Combination Subthresholding approach” consists in showing that a combination of noneffective doses of drugs yields a significant effect, and the “highest single agent approach/cooperative effect” reflects that the resulting effect of a drug combination is greater than the effects produced by its individual components [35]. To determine synergistic CAR activation, we used Bliss independence at the transcription level of the prototypical target gene *Cyp2b10*. The additive effect (E) was calculated as follows: E_TBT+Pesticide_ = E_TBT_ + E_Pesticide_ − E_TBT_ × E_Pesticide_. To apply the model, the results were expressed in % of 2 mg TCPOBOP, representing the maximum effect. For the other data, no maximum effect was available; therefore, the “response additivity” model was used, and the additivity calculated using E_TBT+Pesticide_ = E_TBT_ + E_Pesticide_. The effects were considered synergic when they were significantly superior to the additive effect calculated using a one-sample *t*-test, and *p* < 0.05 was considered significant.

## 3. Results

### 3.1. Modulation of Transcriptional Activity of mCAR by Pesticides

More than 80 pesticides were screened for their ability to activate mCAR. The pesticides were first tested for the non-specific modulation of luciferase expression in the HG5LN parental cell line, which contains the same reporter gene as HG5LN-GAL4-mCAR cells but lacks Gal4-mCAR. Then, the chemicals were tested in HG5LN GAL4-mCAR at concentrations (100 nM to 10 μM) that were not able to inhibit or activate luciferase expression in the HG5LN reporter cell line. The activity of the chemicals in mCAR was compared to the activity of the reference mCAR agonist TCPOBOP. This compound fully activated mCAR, with an EC_50_ of 40 nM (Figure 1).

Six pesticides were selected for the evaluation of mCAR activation in vivo: bifenox, boscalid, bupirimate, dieldrin, pendimethalin, and propiconazole (Figure 1). The strongest activators were propiconazole, bupirimate, and bifenox (Figure 1).

### 3.2. Synergic and Combined Effects on CAR Activation

We then evaluated the co-activation of the CAR-RXR heterodimer by the selected pesticides, ligands of mCAR, in combination with TBT, a known RXR agonist. The activation of the heterodimer was measured by analyzing the expression of the CAR target gene *Cyp2b10.* With the exception of pendimethalin, pesticides alone and TBT alone did not significantly up-regulate *Cyp2b10*. However, when combined with TBT, dieldrin, propiconazole, bifenox, boscalid, and bupirimate up-regulated *Cyp2b10*, demonstrating a combined effect on the activation of the CAR-RXR heterodimer using the combination subthresholding approach (Figure 2). No significant activation was observed for pendimethalin combined with TBT. A synergistic effect was assessed for dieldrin + TBT since the up-regulation of *Cyp2b10* was significantly higher than the calculated Bliss additivity.

### 3.3. NAFLD Induction

This study evaluated the effects of each treatment on the body and liver weights of mice after they had been sacrificed. There was no significant difference in body weight between the different groups; however, the liver/body weight ratio was found to be significantly higher in mice treated with TBT, dieldrin + TBT, propiconazole ± TBT, bifenox ± TBT, boscalid + TBT, bupirimate + TBT, and pendimethalin alone compared to the DMSO group (Table 2). A combined effect was found for bifenox when used in combination with TBT, with its ratio being significantly higher than that of bifenox or TBT alone. Interestingly, the elevated ratio observed with pendimethalin was not observed when pendimethalin was used in combination with TBT (Table 2).

Histological sections of the liver were stained with red oil to test for lipid accumulation (Figure 3). Steatosis was morphologically explored using a well-established scoring system [34] (Table 3). The control (DMSO) did not show any lipid accumulation. Significant steatosis (lipid accumulation) was observed in at least one individual subjected to treatments containing pesticides, excluding bupirimate, and boscalid. Pendimethalin and the combination of TBT with dieldrin, propiconazole, bifenox, boscalid, and bupirimate led to a mean area of lipid droplets greater than 5%, which suggests steatogenic potential for these molecules. All observed steatoses were microvesicular. In some cases, more serious steatosis was observed, characterized by an azonal distribution of lipid droplets, especially in the propiconazole + TBT group. A significant combined effect on lipid droplet area was observed for TBT combined with propiconazole, bifenox, and bupirimate, while the combination of TBT with pendimethalin led to the disappearance of steatosis observed with pendimethalin alone.

Hepatic neutral lipid levels were assessed to quantify the observed steatosis (Table 4). The main components of lipid droplets, cholesterol esters, and TG were quantified. The results showed no effect on cholesterol ester levels. However, a combined effect of TBT on TG accumulation was observed with dieldrin (fold change of 2.43), propiconazole (fold change of 4.96), bifenox (fold change of 8.13), boscalid (fold change of 3.16), and bupirimate (fold change of 4.51). In contrast, pendimethalin lost its increasing effect when combined with TBT (Table 4). These results confirm the observations made of the liver samples (Figure 4). It is worth mentioning that the observed combined effects cannot be defined as synergistic because they were not significantly higher than the calculated additive effect. The observed effects on TG accumulation appear to be additive rather than synergistic.

### 3.4. Changes in Plasmatic Biochemical Profiles

Various liver function markers were measured in each group, including ALAT, ASAT, TG, FFA, total cholesterol, LDL, and HDL levels (Table 5). There was no significant increase in ALAT or ASAT levels when pesticides were used alone or in combination with TBT. However, a slight decrease in ASAT levels was observed for TBT and the propiconazole–TBT mixture. This suggests that despite the observed steatosis, the pesticides and their combinations did not cause significant liver damage, as demonstrated by the liver function markers ALAT and ASAT.

The combination of propiconazole and bifenox with TBT led to a decrease in FFA levels (Table 5). An increase in cholesterol levels was observed in the plasma of animals exposed to TBT in combination with propiconazole or bupirimate. This increase was found to be synergistic for the mixture of propiconazole and TBT since it was significantly higher than the calculated additive effect. This suggests that the combination of propiconazole and TBT leads to a greater increase in cholesterol levels than would be expected if the effects of each substance were simply added together. With the exception of pendimethalin, all the pesticides in combination with TBT increased HDL levels, and a synergistic increase was found for bupirimate. The TG and LDL levels were not significantly altered.

### 3.5. Pesticides Elicit Transcriptional Changes in Key Genes of the Liver’s Metabolic Pathways

To investigate the mechanisms behind the steatosis observed during exposure to pesticides, both alone and in combination with TBT (Figure 3, Table 3 and Table 4), we measured the expression of key genes involved in different hepatic metabolic pathways, such as lipogenesis (fatty acid synthase (*Fasn*) and stearoyl-coenzyme A desaturase 1 (*Scd-1*)), lipid droplet formation (perilipin 3 (*Plin3*)), beta-oxidation (Peroxisomal acyl-coenzyme A oxidase 1 (*Acox1*) and enoyl CoA isomerase (*Eci*)), cholesterol biosynthesis (mevalonate diphosphate decarboxylase (*Mvd*)), fatty acid and cholesterol transport (cluster of differentiation 36 (*CD36*)), and carbohydrate metabolism (glucokinase (*Gck*)). The results of this analysis are shown in Figure 4, which only includes genes that showed effects.

Exposure to propiconazole induced an increase in the expression levels of genes involved in both fatty acid catabolism (*Eci*) and lipogenesis (*Fasn*) when combined with TBT. However, the observed effect was not significantly greater than the calculated theoretical additive effect. The expression of the cholesterol biosynthesis gene *Mvd* was also increased with propiconazole, alone and when combined with TBT. This suggests that a combination of propiconazole and TBT leads to an increase in the expression of genes involved in both fatty acid catabolism and lipogenesis, but this increase is not greater than would be expected if the effects of each substance were simply added together (Figure 4A).

Exposure to bifenox alone induced the dysregulation of several enzymes involved in fatty acid catabolism, such as *Acox1* and *Eci*, fatty acid transporter *CD36*, glucokinase *Gck*, and cholesterol biosynthesis gene *Mvd*. When bifenox was combined with TBT, there was synergistic activation of *CD36* at the transcriptional level, which was higher than what would be expected from the effects of the two chemicals alone or from the calculated additive effect (Figure 4B). Additionally, when bifenox was combined with TBT, the expression of *Gck* was not further increased compared to when exposed to bifenox alone.

Exposure to bupirimate induced the up-regulation of *Fasn* when administered alone, and of *Scd1*, *Plin3*, and *Mvd* when administered in combination with TBT (Figure 4C). The effect on *Mvd* was confirmed to be synergistic, as the observed increase in its expression was significantly higher than the additive individual effects of bupirimate and TBT (Figure 4C).

Exposure to pendimethalin alone increased the expression of *Plin3*, but when pendimethalin was combined with TBT, its effect on Plin3 expression was lost (Figure 5).
Figure 4Gene expression levels of key metabolic pathway genes in the livers of mice according to RT-qPCR. Key genes involved lipogenesis (*Fasn* and *Scd-1*), lipid droplet formation (*Plin3*), beta-oxidation (*Acox1* and *Eci*), cholesterol biosynthesis (*Mvd*), fatty acid and cholesterol transport (*CD36*), and carbohydrate metabolism (*Gck*), deregulated by exposure to propiconazole (**A**), bifenox (**B**), bupirimate (**C**), and pendimethalin (**D**). The graph shows the results of the fold changes of the DMSO group. Data are presented as mean ± standard error. * *p* < 0.05, ** *p* < 0.01, *** *p* < 0.001, **** *p* < 0.0001. *p*-values represent significant differences between pesticide-treated mice and DMSO-treated mice according to one-way ANOVA test followed by Tukey’s multiple comparisons test. # *p* < 0.05, ## *p* < 0.01 represents significant difference between the observed effect of the combination of pesticide with TBT and the calculated response additivity of this combination according to a one-sample *t*-test.
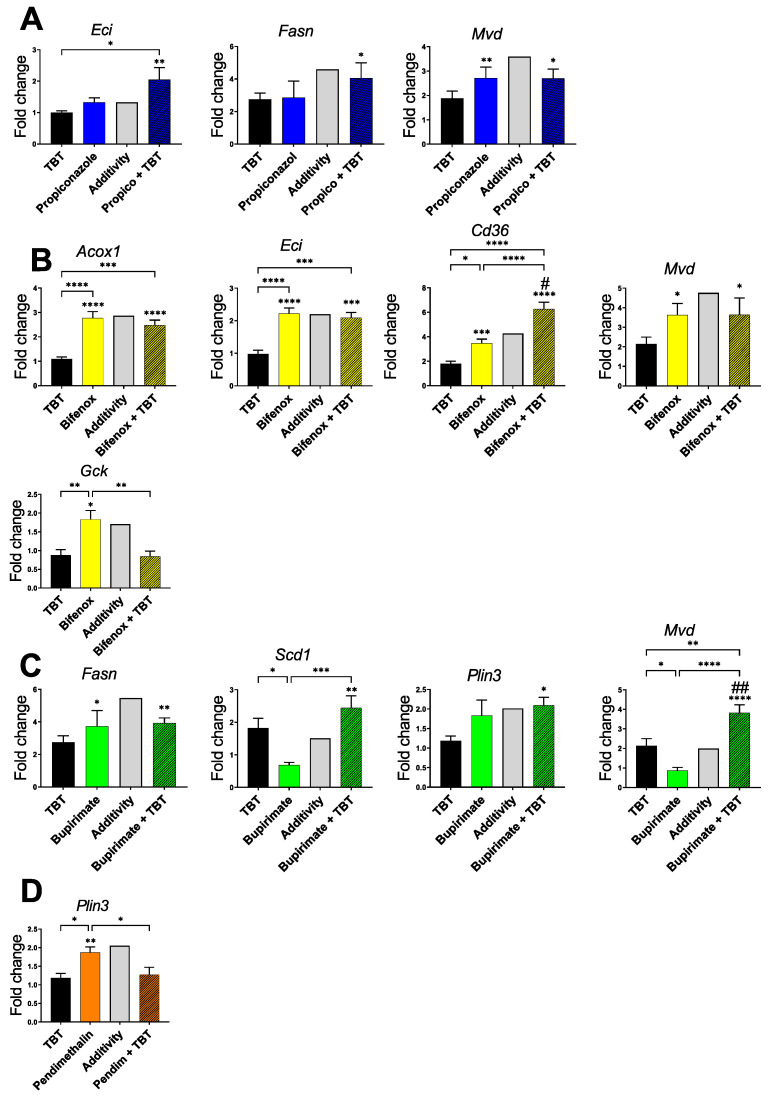


## 4. Discussion

In this study we selected six pesticide activators of the CAR based on screening carried out in vitro. The results of the screening indicated that propiconazole, bupirimate, and bifenox were more effective CAR activators than dieldrin, boscalid, and pendimethalin, and were found to be as effective at activating CAR as the pharmacological CAR agonist TCPOBOP.

One of the most important results of this study is the demonstration of a combined, sometimes synergistic effect of CAR-RXR heterodimer activation. Typically, when studying the activation of nuclear receptors by xenobiotics, many studies tend to concentrate on the activation of a single receptor, such as CAR or PXR, by a single molecule [19,36]. A study by Delfosse et al. highlighted the importance of considering the combined effect of multiple environmental contaminants on nuclear receptors, particularly on PXR [26]. They demonstrated the formation of “supramolecular ligands” within the ligand-binding pocket of PXR, which contribute to the synergistic toxic effects of chemical mixtures. Moreover, in their latest study, Delfosse et al. demonstrated that the synergistic activation of PXR could involve not only the binding of several molecules to the ligand pocket of the receptor, but also the coactivation of the two partners of the PXR–RXR heterodimer [25]. This means that PXR and RXR environmental ligands can act together to induce cooperative recruitment of the coactivator steroid receptor coactivator-1 (SRC-1) by the heterodimer, and the synergistic activation of PXR–RXR target gene expression. The originality of the present study lies in its demonstration that such synergistic effects of multiple environmental contaminants on nuclear receptors occur not only in vitro, but also in vivo, and with another nuclear receptor, CAR, in addition to PXR. We demonstrate that activating CAR using a mixture of dieldrin and TBT has a synergistic effect, and using mixtures of propiconazole, bifenox, boscalid, and bupirimate with TBT has additive effects.

Our findings are relevant to the idea of heterodimer permissiveness with RXR. Previous studies have distinguished two categories of nuclear receptor heterodimer: permissive and nonpermissive [37]. Permissive heterodimers, such as RXR/PPARg, RXR/LXR, or RXR/FXR, can be activated by ligands of either RXR or its partner receptor, and their activation is synergistically enhanced when both ligands are present [38]. On the other hand, nonpermissive heterodimers, such as RXR/TR or RXR/VDR, cannot be activated by ligands that bind to RXR, making RXR a “silent” partner in these cases [38]. However, the permissiveness or non-permissiveness of a given heterodimer can depend on various factors, such as the specific ligands, DNA sequences, cellular environment, and post-translational modifications involved [39]. The CAR/RXR heterodimer has been shown to be permissive or non-permissive, depending on the cellular context and the CAR isoform [40,41]. Our results suggest permissiveness of the CAR/RXR complex with the activation of RXR by TBT.

One notable feature of CAR is its constitutive activity, meaning that it can activate target gene expression in the absence of ligand binding [41]. The activation of CAR by a xenobiotic releases it from its cytoplasmic retention complex, allowing it to migrate to the nucleus, where it can activate the expression of its prototypical target genes [42]. Further research is needed to determine how RXR activation by TBT takes place within the CAR/RXR complex in such a configuration.

The development of steatosis, or fatty liver, can occur through various mechanisms, such as an increase in lipid synthesis, a decrease in lipid catabolism, enhanced lipid uptake or a reduction in lipid export [43]. Analyses of genes related to these metabolic pathways using RT-qPCR have provided insights into the mechanisms of lipid accumulation induced by the combination of TBT with pesticides. The results of this analysis show that exposure to propiconazole, bifenox and pendimethalin can lead to an increase in the expression of lipogenic genes, indicating a possible role of this pathway in the formation of steatosis. Bifenox induces lipid catabolism through the up-regulation of *Eci* and *Acox1*, as well as lipid uptake through up-regulation of the *CD36* transporter. In this case, steatosis likely results from the overwhelmed liver’s ability to catabolize fatty acids through an increase in their uptake. This theory is supported by the observation of a decrease in free fatty acids in the blood of mice exposed to a mixture of bifenox and TBT (Table 5), which suggests that these fatty acids are taken up by the liver through the *CD36* transporter (Figure 5). These results are in agreement with previous studies in which a steatogenic effect of pesticides occurs via similar or different mechanisms [17,44].

We also revealed the effect of exposure to the mixture of TBT and pesticides on dyslipidemia (Table 5). The combination of propiconazole with TBT increased cholesterol levels, and with the exception of pendimethalin, all the pesticides combined with TBT increased HDL levels. Interestingly, the observed effects occurred for certain combinations (TBT with propiconazole or bupirimate) of the synergistic type. The occurrence of lipid metabolism disorders caused by the combination of TBT with propiconazole, bifenox, or bupirimate may be partly due to the up-regulation of the *Mvd* gene, which is involved in cholesterol synthesis in the liver (Figure 5). However, those induced via the combination of TBT with dieldrin or boscalid may involve other mechanisms. With RXR, such as CAR, being involved in the pathogenesis of metabolic syndrome, the pan-activation of the permissive CAR/RXR heterodimer amplifies metabolic disorders induced by xenobiotics, as underlined by other authors [45].

The combination of pesticides with TBT results in greater production of lipids than either substance alone, with the effects being additive rather than synergistic. The combination of TBT with dieldrin, propiconazole, bifenox, or bupirimate results in concomitant lipid accumulation (Figure 3 and Table 3) and CAR activation (Figure 2). Moreover, the combination of TBT with pendimethalin leads to a decrease in both lipid accumulation (Figure 3 and Table 3) and CAR activation (Figure 2) compared to pendimethalin alone. These results indicate the potential involvement of the CAR receptor in these effects, which is consistent with other studies reporting on the role of CAR in NAFLD induction or progression [22,46].

Our results also raise the question: why does the liver produce fat when it is exposed to certain toxins? Several studies have described an accumulation of lipophilic toxins in adipose tissue [47]. This accumulation represents a means of neutralizing these toxins, preventing them from exerting their toxicity on other tissues. Our findings raise the possibility that the lipid droplets produced by the liver may be designed to capture toxic compounds until they can be eliminated from the body. This lipid production could form part of a detoxification process, through which the liver could neutralize toxins by trapping them in lipid droplets while awaiting their elimination via lipophagy [48].

## 5. Conclusions

In summary, this article shows, for the first time, a synergistic increase in the gene transcription of *Cyp2b10*, a target of CAR, and of genes involved in the regulation of the liver lipid metabolism by mixtures of pollutants in vivo. It also highlights the effects of this activation on the disruption of liver and plasma lipids. These observations on short-term exposure highlight the need for further research on the long-term effects of low-level exposure to these pesticides. This research contributes to the growing understanding of how environmental contaminants can influence nuclear receptor activity and the associated health risks.

## Figures and Tables

**Figure 1 cells-12-01201-f001:**
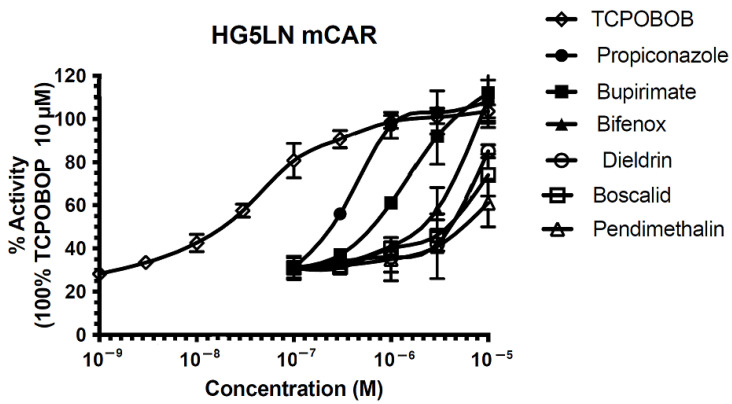
The graph shows mCAR’s transcriptional activity in response to the CAR agonist TCPOBOP and various pesticides. The results are expressed as a percentage of the highest luciferase activity triggered by 10 μM TCPOBOP. Standard deviations are represented by error bars.

**Figure 2 cells-12-01201-f002:**
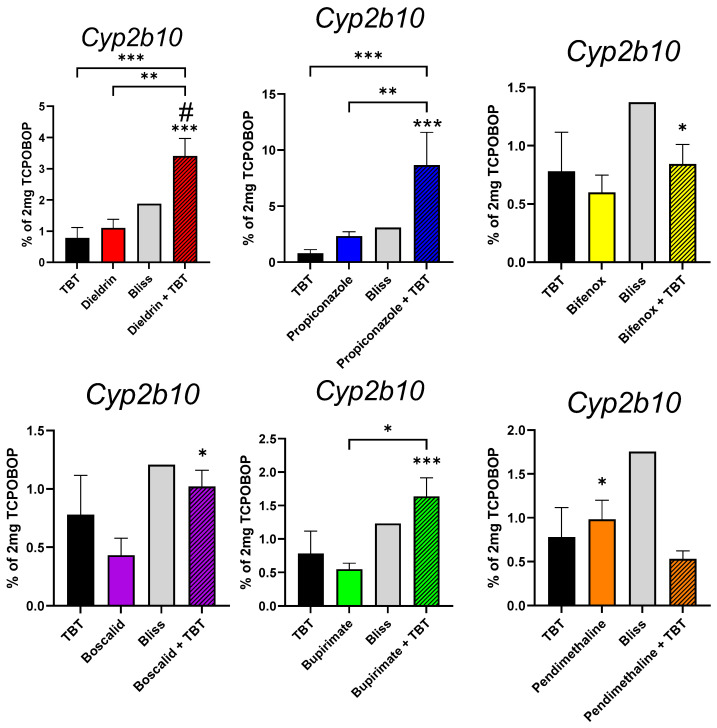
Gene expression levels of the CAR prototypical target gene, *Cyp2b10*, in the livers of mice according to RT-qPCR. The results are presented as a graph showing the expression levels in % of the 2 mg TCPOBOP (100%) and DMSO groups (0%). The data are presented as mean ± standard error of the means. The statistical analysis used a one-way ANOVA test followed by Tukey’s multiple comparisons test to determine the difference between the pesticide-treated mice and DMSO-treated mice. The *p*-values indicate the level of significance, with * *p* < 0.05, ** *p* < 0.01, and *** *p* < 0.001 indicating significant differences. Additionally, the study also used a one-sample *t*-test to determine the difference between the observed effect of TBT + pesticide and the calculated Bliss additivity of TBT + pesticide, represented by # *p* < 0.05.

**Figure 3 cells-12-01201-f003:**
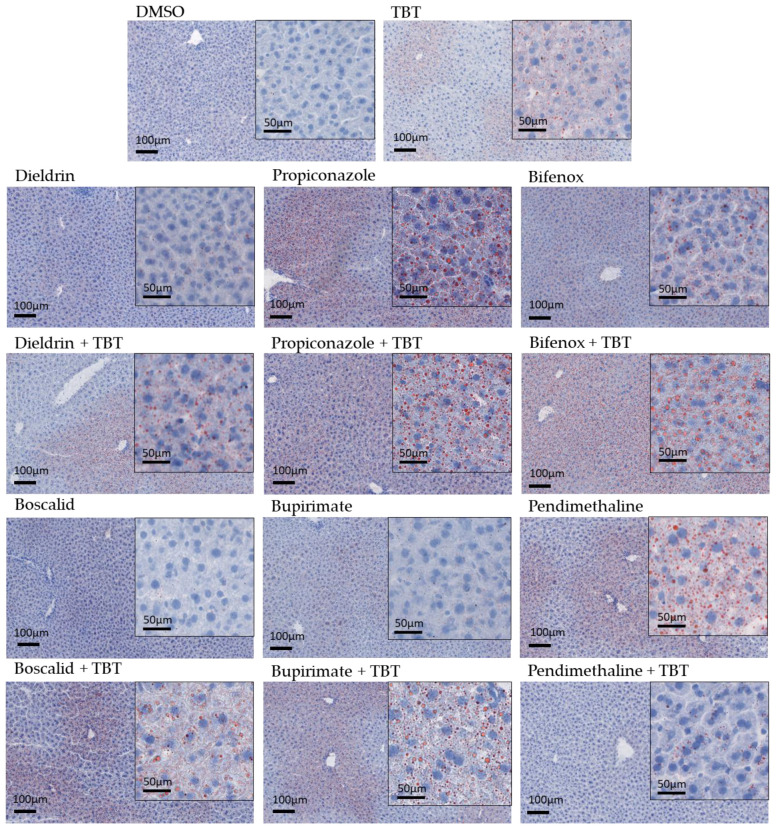
Histologic hepatic steatosis assessment was conducted on frozen liver sections stained with Harris Hematoxylin and red oil stain. Hematoxylin-highlighted nuclei in blue; red oil-stained lipid droplets in red. An accumulation of multiple small red droplets in cytoplasm contributed to steatosis. Zoom focus: 20× and 40×.

**Figure 5 cells-12-01201-f005:**
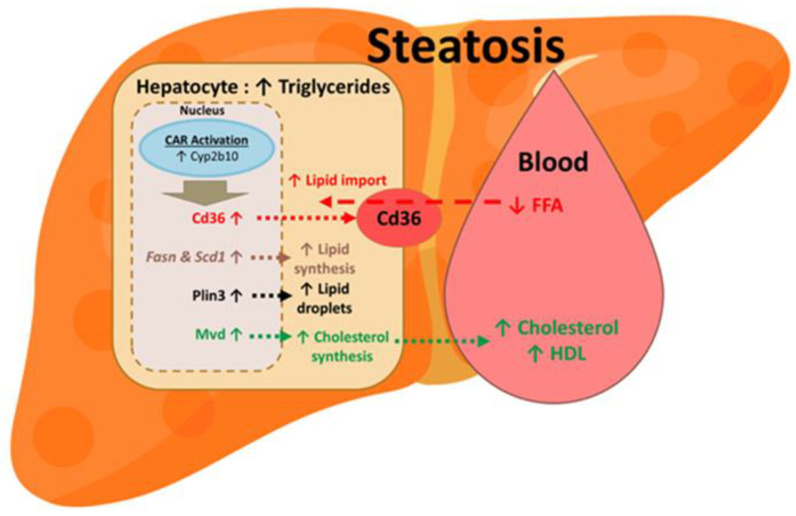
Suggested mechanisms of steatosis and metabolic disruption induced by pesticides. Constitutive androstane receptor (CAR), fatty acid transporter (*Cd36*), fatty acid synthase (*Fasn*), stearoyl-CoA desaturase (*Scd1*), perilipin 3 (Plin3), mevalonate diphosphate decarboxylase (*Mvd*), FFA, and HDL.

**Table 1 cells-12-01201-t001:** Sequence of the primers used in RT-qPCR.

Gene	Primer Sequence F 5’-3’	Primer Sequence R 3’-5’
*Acox1*	AGACCCTGAAGAAATCATGTGG	AGGAACATGCCCAAGTGAAG
*Cd36*	GTTAAACAAAGAGGTCCTTACACATACAG	AGTGAAGGCTCAAAGATGGC
*Cyp2b10*	TTTCTGCCCTTCTCAACAGGAA	TGGACGTGAAGAAAAGGAACAAC
*Eci*	GTTCACCATCAGCCTGGAGAAG	AGAAGATACCCGGGCATTCC
*Fasn*	AGTCAGCTATGAAGCAATTGTGGA	CACCCAGACGCCAGTGTTC
*Gck*	TCGCAGGTGGAGAGCGA	TCGCAGTCGGCGACAGA
*Mvd*	CGGTCAACATCGCAGTTATCAA	GTGCAGCGTGACGCTCAG
*Plin3*	GGCTGGACAGACTGCAGGA	TCTTGAGCCCCAGACACTGTAG
*Scd1*	CAGTGCCGCGCATCTCTAT	CAGCGGTACTCACTGGCAGA
*Tbp*	ACTTCGTGCAAGAAATGCTGAA	GCAGTTGTCCGTGGCTCTCT

**Table 2 cells-12-01201-t002:** Effect of pesticide treatment on liver/body weight. The liver-to-body weight ratios expressed in fold change of the DMSO group, (mean ± standard error). “a” indicates significant difference with DMSO, “b” with TBT, and “c” with pesticide treatment alone. *p* < 0.05 is considered significant according to one-way ANOVA test followed by Tukey’s multiple comparisons test.

Treatment	Liver/Body Weight
TBT	1.33 ± 0.06 ^a^
Dieldrin	1.06 ± 0.05
Dieldrin + TBT	1.17 ± 0.10 ^a^
Propiconazole	1.23 ± 0.05 ^a^
Propiconazole + TBT	1.47 ± 0.09 ^a^
Bifenox	1.19 ± 0.05 ^a^
Bifenox + TBT	1.46 ± 0.08 ^abc^
Boscalid	1.11 ± 0.12
Boscalid + TBT	1.25 ± 0.15 ^a^
Bupirimate	1.04 ± 0.15
Bupirimate + TBT	1.34 ± 0.09 ^a^
Pendimethalin	1.29 ± 0.09 ^a^
Pendimethalin + TBT	1.10 ± 0.09

**Table 3 cells-12-01201-t003:** Morphological assessment and scoring of steatoses. Data are presented as number of individuals per stage, per zone, per microvesicular morphology. Steatosis degrees: <5%; 5% ≤ Stage 1 < 33%. Area of lipid droplets is presented as mean ± standard error. “a” indicates significant difference with DMSO. *p* < 0.05 is considered significant according to one-way ANOVA test followed by Tukey’s multiple comparisons test.

Treatment	Steatosis Degree			Location		Microvesicular Morphology
	<5%	Stage 1	Mean Area %	Zone 3	Azonal	
DMSO	6	0	0.50 ± 0.43	0	0	0
TBT	5	1	1.95 ± 2.19	1	0	1
Dieldrin	3	3	4.11 ± 2.62	3	0	3
Dieldrin + TBT	5	1	5.27 ± 7.59	1	0	1
Propiconazole	2	4	6.51 ± 2.51	3	1	4
Propiconazole + TBT	2	3	12.81 ± 9.79 ^a^	0	3	3
Bifenox	3	3	5.91 ± 3.56	2	1	3
Bifenox + TBT	3	3	6.49 ± 5.99 ^a^	2	1	3
Boscalid	3	0	2.08 ± 1.58	0	1	1
Boscalid + TBT	4	2	4.19 ± 4.76	2	0	1
Bupirimate	6	0	1.33 ± 0.71	0	0	0
Bupirimate + TBT	3	3	7.31 ± 7.85 ^a^	3	0	3
Pendimethalin	2	4	5.78 ± 3.00 ^a^	4	0	4
Pendimethalin + TBT	5	1	2.14 ± 1.72	1	0	1

**Table 4 cells-12-01201-t004:** Effects of pesticide treatments on triglycerides and cholesterol esters. Triglycerides and cholesterol esters expressed in fold change of DMSO group (mean ± standard error). “a” indicates significant difference with DMSO. *p* < 0.05 is considered significant according to one-way ANOVA test followed by Tukey’s multiple comparisons test.

Treatment	Triglycerides	Cholesterol Esters
TBT	1.52 ± 0.74	0.57 ± 0.13
Dieldrin	1.53 ± 0.60	1.08 ± 0.23
Dieldrin + TBT	2.43 ± 4.51 ^a^	1.03 ± 0.47
Propiconazole	2.79 ± 0.72	1.19 ± 0.43
Propiconazole + TBT	4.96 ± 2.68 ^a^	1.38 ± 0.10
Bifenox	5.47 ± 1.65	1.48 ± 0.23
Bifenox + TBT	8.13 ± 4.74 ^a^	1.11 ± 0.57
Boscalid	2.10 ± 1.05	1.33 ± 0.73
Boscalid + TBT	3.16 ± 1.40 ^a^	0.75 ± 0.27
Bupirimate	2.76 ± 1.19	0.98 ± 0.14
Bupirimate + TBT	4.51 ± 1.61 ^a^	1.04 ± 0.42
Pendimethalin	3.36 ± 1.55 ^a^	0.86 ± 0.18
Pendimethalin + TBT	2.66 ± 1.45	0.95 ± 0.07

**Table 5 cells-12-01201-t005:** Effects of pesticide treatments on plasmatic biochemical parameters. Results are expressed as fold change of DMSO group (mean ± standard error). “a” represents significant difference with DMSO. *p* < 0.05 is considered significant according to one-way ANOVA test followed by Tukey’s multiple comparisons test. # indicates a significant difference (*p* < 0.05) between the observed effect of TBT + Pesticide and the calculated additivity of TBT + Pesticide using a one-sample *t*-test.

Treatment	TG	FFA	Cholesterol	HDL	LDL
TBT	1.17 ± 0.24	0.69 ± 0.20	1.12 ± 0.08	1.19 ± 0.1	1.01 ± 0.27
Dieldrin	1.47 ± 0.62	1.05 ± 0.40	1.17 ± 0.19	1.16 ± 0.13	1.11 ± 0.30
Additivity	1.64	0.73	1.29	1.35	1.12
Dieldrin + TBT	1.21 ± 0.38	0.90 ± 0.11	1.26 ± 0.18	1.29 ± 0.13 ^a^	1.4 ± 0.36
Propiconazole	1.06 ± 0.14	0.91 ± 0.29	1.07 ± 0.14	1.12 ± 0.13	0.88 ± 0.31
Additivity	1.23	0.6	1.19	1.31	0.89
Propiconazole + TBT	1.16 ± 0.23	0.54 ± 0.07 ^a^	1.31 ± 0.10 ^a#^	1.34 ± 0.14 ^a^	1.34 ± 0.78
Bifenox	0.87 ± 0.23	0.89 ± 0.30	1.21 ± 0.09	1.23 ± 0.14	0.94 ± 0.12
Additivity	1.04	0.57	1.33	1.42	0.95
Bifenox + TBT	0.81 ± 0.33	0.77 ± 0.17 ^a^	1.21 ± 0.09	1.27 ± 0.10 ^a^	0.8 ± 0.28
Boscalid	1.27 ± 0.33	1.01 ± 0.19	1.03 ± 0.18	1.11 ± 0.20	0.86 ± 0.12
Additivity	1.44	0.7	1.15	1.31	0.87
Boscalid + TBT	1.51 ± 0.60	0.95 ± 0.26	1.24 ± 0.06 ^#^	1.3 ± 0.09 ^a^	1.02 ± 0.13
Bupirimate	1.32 ± 0.55	1.23 ± 0.26	1.09 ± 0.16	1.09 ± 0.21	1.14 ± 0.20
Additivity	1.49	0.92	1.21	1.28	1.15
Bupirimate + TBT	1.16 ± 0.32	0.84 ± 0.21	1.35 ± 0.09 ^a^	1.36 ± 0.07 ^a#^	1.24 ± 0.15
Pendimethalin	1.12 ± 0.53	0.84 ± 0.34	1.24 ± 0.07	1.34 ± 0.08	1.08 ± 0.37
Additivity	1.29	0.52	1.36	1.54	1.09
Pendimethalin + TBT	1.09 ± 0.63	1.01 ± 0.51	0.9 ± 0.39	0.94 ± 0.42	0.96 ± 0.52

## Data Availability

The data presented in this study are available on request from the corresponding author.

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
