# Peer review of "Steatosis and Metabolic Disorders Associated with Synergistic Activation of the CAR/RXR Heterodimer by Pesticides"

_cells, 2023, doi:10.3390/cells12081201_

Round 1

Reviewer 1 Report

Article title: Synergic activation of the CAR/RXR heterodimer by pesticides and induced metabolic effects.

Journal: Cells

Article ID: cells-2223423

Overview:

The current study aimed to evaluate whether a synergistic activation of nuclear receptors by environmental pollutants can occur in vivo and to assess the consequences on the development of metabolic disorders.

Identification of new ways for an opportune diagnosis of metabolic diseases as non-alcoholic fatty liver disease (NAFLD) are required to avoid the progression to an advanced stage as liver fibrosis, cirrhosis, and hepatic carcinoma. The approach of this research study is the evaluation of the relationship between the modification of markers of metabolic disorders related with lipogenesis through the activation of nuclear factors by six pesticides and the development of Toxic Associated Fatty Liver Disease (TAFLD).

This article was previously revised, it is overall well-presented and describes original research in the area of hepatotoxicity (cellular steatosis) related with the exposure to xenobiotics such as pesticides and molecular diagnosis. However, some parts of the manuscript still need to be improved.

Based on the above mentioned, I found the present research study adequate to be accepted and I suggest Minor Revisions to be considered for publication at Cells.

Comments:

·      Title:

1.     Title needs to be rewrite as “Liver steatosis induced by pesticides and the metabolic effects associated with the CAR/RXR heterodimer-related markers of activation”, since there is no direct evidence of the activation by Western blot or qPCR of CAR/RXR heterodimer, authors only evaluated genes related with but not the nuclear factors per se.

·      Introduction:

1.    Introduction section provides enough information about the state of art of non-alcoholic fatty liver diseases related with xenobiotics. However, authors should detailed describe the mechanisms and diagnosis of hepatotoxicity (steatogenic effect) related with the exposure to environmental toxins used in their study from the same cited references or including more recent references (Liebe, R., Esposito, I., Bock, H. H., vom Dahl, S., Stindt, J., Baumann, U., Luedde, T., & Keitel, V. (2021). Diagnosis and management of secondary causes of steatohepatitis. Journal of Hepatology, 74(6), 1455-1471. https://doi.org/10.1016/j.jhep.2021.01.045).

2.    Correct a double-space in line 48.

·      Materials:

1.     Include version of the software used (GraphPad Prism) in line 117 as in line 188.

2.     Authors describe that animal experiments were conducted following the European guidelines and were approved by an independent Ethical committee, but they must include a code or number for both.

3.       Include a space between the words “Tukey’s multiple” in line 189.

·      Results:

1.     In general, results are clear and described in an appropriate way.

2.     In line 224 phrases “in vivo”, “in mice” is redundant since authors already described that they used mice in Materials section.

3.     In figure 2, authors must include microscale at each micrograph.

4.     In line 273, include capital letter in word “morphological” of table 3.

5.     Line 287, correct the word “steatoses”.

6.     ¿Why did the authors not include the results of the main biochemical markers of liver damage ALT and AST?

7.     Authors must make the name of the pesticides used homogeneous in the whole document since some times they are with small letters and most of the times they initiate with capital letter (eg. Lines 283-284, 374-376, 379-380).

8.     In lines 333-334, authors must define all initials or no one.

9.     ¿May the authors could explain why they did not include Western Blot or qPCR analysis of the nuclear factors CAR and RXR to undergo, explain and directly relate their expression or modulation with the alterations of the markers of liver damage and lipogenesis in this study?

·      Discussion:

1.     Lines 409-410, phrase “studies that aim to study” is redundant.

2.     Discussion section only have 8 cites of 38 total cites of the manuscript when it must be strong section with at least 1/3 of the whole references cited in the document.

3.     Authors need to rewrite this section to improve it since they describe mostly parts of the results but they do not compare and describe how their findings are related with previous studies. Discussion section is too weak.

4.     Authors need to detail describe the mechanism or cascade of events that involve the activation of the CAR/RXR heterodimer and how they induce the modulation of different enzymes as response against the exposure to environmental toxics as pesticides, and to explain how their findings correlate with the steatosis and fatty liver disease.

5.     I suggest the following references that may improve the discussion section:

o   Xu, R. X., Lambert, M. H., Wisely, B. B., Warren, E. N., Weinert, E. E., Waitt, G. M., Williams, J. D., Collins, J. L., Moore, L. B., Willson, T. M., & Moore, J. T. (2004). A Structural Basis for Constitutive Activity in the Human CAR/RXRα Heterodimer. Molecular Cell, 16(6), 919-928. https://doi.org/10.1016/j.molcel.2004.11.042

o   Shizu R, Benoki S, Numakura Y, Kodama S, Miyata M, et al. (2013) Xenobiotic-Induced Hepatocyte Proliferation Associated with Constitutive Active/Androstane Receptor (CAR) or Peroxisome Proliferator-Activated Receptor α (PPARα) Is Enhanced by Pregnane X Receptor (PXR) Activation in Mice. PLOS ONE 8(4): e61802. https://doi.org/10.1371/journal.pone.0061802

o   Shulman, A. I., & Mangelsdorf, D. J. (2005). Retinoid X Receptor Heterodimers in the Metabolic Syndrome. New England Journal of Medicine, 353(6), 604-615. https://doi.org/10.1056/NEJMra043590

o   Kojetin, D., Matta-Camacho, E., Hughes, T. et al. Structural mechanism for signal transduction in RXR nuclear receptor heterodimers. Nat Commun 6, 8013 (2015). https://doi.org/10.1038/ncomms9013

6.     Authors suggest in the results section that ALT and AST were not significantly altered.

¿May the authors can explain with scientific evidence why the levels of the main biochemical markers of liver damage ALT and AST were not altered?

·      Conclusions:

This section needs to be included separately since it is already at the Discussion section and represents a reasonable part of it (eg. 464-480).

I highly recommend to include an illustration (figure) at this section of the suggested molecular mechanism of action induced by pesticides at hepatocytes level and their metabolic effect in the liver.

Author Response

We thank the reviewer for his suggestions which improved the quality of the manuscript.

  1. Title needs to be rewrite as “Liver steatosis induced by pesticides and the metabolic effects associated with the CAR/RXR heterodimer-related markers of activation”, since there is no direct evidence of the activation by Western blot or qPCR of CAR/RXR heterodimer, authors only evaluated genes related with but not the nuclear factors per se.

#authors:  We thank the reviewer and accept the suggested title which is more precise. But we would like to add “synergic”. The rewritten title is “Liver steatosis induced by pesticides and the metabolic effects associated with the synergic induction of CAR/RXR heterodimer-related markers of activation” The title has been modified in the new manuscript. Indeed, we used the CAR activation marker Cyp2b10 because CAR does not autoinduce its own transcription when activated. Therefore, the transcription level of CAR and it’s proteic levels observed in western blotting would not be the most relevant when studying the activation of this nuclear receptor. Western blot would give us information on CAR induction but not on CAR activation.

  • Introduction:
  1. Introduction section provides enough information about the state of art of non-alcoholic fatty liver diseases related with xenobiotics. However, authors should detailed describe the mechanisms and diagnosis of hepatotoxicity (steatogenic effect) related with the exposure to environmental toxins used in their study from the same cited references or including more recent references (Liebe, R., Esposito, I., Bock, H. H., vom Dahl, S., Stindt, J., Baumann, U., Luedde, T., & Keitel, V. (2021). Diagnosis and management of secondary causes of steatohepatitis. Journal of Hepatology, 74(6), 1455-1471. https://doi.org/10.1016/j.jhep.2021.01.045).

#authors:  We thank the reviewer for raising this issue.

We included the suggested reference and developed the description of diagnosis of NAFLD : During a diagnosis of steatohepatitis, once alcoholism and metabolic syndrome have been excluded secondary causes must be explored. Various factors can contribute to NAFLD such as infections (hepatitis C virus), endocrine disorders (hypothyroidism, hypopituitarism), coeliac disease, genetic diseases (Wilson desease, A1AT deficiency), drugs (methotrexate, tamoxifen, corticosteroids…) but also environmental contaminants.

We further developed in the introduction the mechanisms of NAFLD: Indeed, the activation of hepatic nuclear receptors by pollutants is known to be associated with mechanisms contributing to steatosis such as increasing de novo lipogenesis, decreased fatty acid oxidation, increased hepatic lipid uptake, decreased gluconeogensesis.

  1. Correct a double-space in line 48.

 #authors:   This has been done.

  • Materials:
  1. Include version of the software used (GraphPad Prism) in line 117 as in line 188.

#authors:   Included:  GraphPad prism 9.

  1. Authors describe that animal experiments were conducted following the European guidelines and were approved by an independent Ethical committee, but they must include a code or number for both.

#authors:  Included: An independent ethics committee (Toxcométhique, INRAE ToxAlim, Toulouse, France) approved the experiment (Approval Code: Toxcom247)

  1. Include a space between the words “Tukey’s multiple” in line 189.

#authors:  This has been done.

  • Results:
  1. In general, results are clear and described in an appropriate way.
  2. In line 224 phrases “in vivo”, “in mice” is redundant since authors already described that they used mice in Materials section.

#authors:  the redundant “In vivo” is deleted from the text

  1. In figure 2, authors must include microscale at each micrograph.

#authors: This has been done

  1. In line 273, include capital letter in word “morphological” of table 3.

#authors:  This has been done

  1. Line 287, correct the word “steatoses”.

#authors:  This has been done

  1. ¿Why did the authors not include the results of the main biochemical markers of liver damage ALT and AST?

#authors:  We thank the reviewer for this interesting question. There was no significant increase in ALT and AST levels, a result in favor of the absence of hepatic lysis. Indeed, this is important information mentioned in the text. But, due to the important data sets already present in this article we decided to leave out the data sets of ALT and AST, showing no liver damage.

  1. Authors must make the name of the pesticides used homogeneous in the whole document since sometimes they are with small letters and most of the times they initiate with capital letter (eg. Lines 283-284, 374-376, 379-380).

#authors:  This issue has been corrected for all pesticides. As they are noncommercial chemical names we wrote the names with lowercase letters as expected.

  1. In lines 333-334, authors must define all initials or no one.

#authors:  All initials have been defined now.

  1. ¿May the authors could explain why they did not include Western Blot or qPCR analysis of the nuclear factors CAR and RXR to undergo, explain and directly relate their expression or modulation with the alterations of the markers of liver damage and lipogenesis in this study?

#authors:  This is an interesting question from the reviewer. We used the CAR activation marker Cyp2b10 because CAR does not autoinduce its own transcription when activated. Therefore, the transcription level of CAR and it’s proteic levels observed in western blotting would not be the most relevant for us because we mainly focus on studying the activation of this nuclear receptor. Western blot would give us information on CAR induction but not on CAR activation.

  • Discussion:
  1. Lines 409-410, phrase “studies that aim to study” is redundant.

#authors:  The sentence was replaced by : “Typically, when studying the activation of nuclear receptors by xenobiotics, many studies tend to concentrate on the activation the activation of a single receptor, such as CAR or Pregnane X Receptor (PXR), by a single molecule”.

  1. Discussion section only have 8 cites of 38 total cites of the manuscript when it must be strong section with at least 1/3 of the whole references cited in the document.

#authors:  More references have been introduced into the discussion section.

  1. Authors need to rewrite this section to improve it since they describe mostly parts of the results but they do not compare and describe how their findings are related with previous studies. Discussion section is too weak.

#authors: We thank the reviewer for his suggestion. Excessively detailed descriptions of results have been removed. Elements have been added to the discussion to enrich it.

  1. Authors need to detail describe the mechanism or cascade of events that involve the activation of the CAR/RXR heterodimer and how they induce the modulation of different enzymes as response against the exposure to environmental toxics as pesticides, and to explain how their findings correlate with the steatosis and fatty liver disease.

#authors: A section has been added in the discussion on this subject and a figure summarizing the mechanisms of steatosis induced by pesticides has been added (Figure 5).

  1. I suggest the following references that may improve the discussion section:

o   Xu, R. X., Lambert, M. H., Wisely, B. B., Warren, E. N., Weinert, E. E., Waitt, G. M., Williams, J. D., Collins, J. L., Moore, L. B., Willson, T. M., & Moore, J. T. (2004). A Structural Basis for Constitutive Activity in the Human CAR/RXRα Heterodimer. Molecular Cell, 16(6), 919-928. https://doi.org/10.1016/j.molcel.2004.11.042

o   Shizu R, Benoki S, Numakura Y, Kodama S, Miyata M, et al. (2013) Xenobiotic-Induced Hepatocyte Proliferation Associated with Constitutive Active/Androstane Receptor (CAR) or Peroxisome Proliferator-Activated Receptor α (PPARα) Is Enhanced by Pregnane X Receptor (PXR) Activation in Mice. PLOS ONE 8(4): e61802. https://doi.org/10.1371/journal.pone.0061802

Shulman, A. I., & Mangelsdorf, D. J. (2005). Retinoid X Receptor Heterodimers in the Metabolic Syndrome. New England Journal of Medicine, 353(6), 604-615. https://doi.org/10.1056/NEJMra043590

Kojetin, D., Matta-Camacho, E., Hughes, T. et al. Structural mechanism for signal transduction in RXR nuclear receptor heterodimers. Nat Commun 6, 8013 (2015). https://doi.org/10.1038/ncomms9013

#authors: We thank the reviewer for his suggestion. The discussion part has been enriched particularly with respect to the notion of permissiveness of the CAR/RXR complex and the constitutively active aspect of the CAR receptor. References suggested by the reviewer have been added.

  1. Authors suggest in the results section that ALT and AST were not significantly altered.

¿May the authors can explain with scientific evidence why the levels of the main biochemical markers of liver damage ALT and AST were not altered?

#authors: Exposure for 3 days to the pesticides alone or in combination with TBT led to the accumulation of lipids in the liver but not to an increase in the levels of ALT and AST, suggesting that this benign steatosis was not deleterious for the liver. It would probably have become so if the exposition had been maintained longer.

Conclusions: This section needs to be included separately since it is already at the Discussion section and represents a reasonable part of it (eg. 464-480).

#authors: A conclusion section has been added

I highly recommend to include an illustration (figure) at this section of the suggested molecular mechanism of action induced by pesticides at hepatocytes level and their metabolic effect in the liver.

#authors: A figure has been added to illustrate the mechanisms by which pesticides induce metabolic disorders.

Reviewer 2 Report

Dauwe et al. reported that activation of CAR/RXR heterodimer by pesticides caused hepatic lipids accumulation. The pesticides treatment regulated de novo lipogenesis- and lipolysis-related genes expression. The concept of this study is interesting. However, there are some issues need to be addressed.

1. The results of study revealed that the pesticides induced simple steatosis but did not significantly affect the dysfunction of the liver. In addition, the impacts of pesticides on the inflammation of liver are not known. I think 3 days is too short a time for pesticides treatment. Therefore, I suggest that prolonging the duration of drug treatment has yielded a more precise conclusion. I would also like to know the effect of pesticides on the body weight of mice.

2. The authors should explain why these pesticides doses selected. How does the doses of these pesticides used in the current study compare with the residual in the environment?

3. With the exception of Figure 3, the presentation of the results is confusing. I suggest that the tables and bar graphs also need to display as Figure 3 (i.e. clearly indicated the negative and/or positive control group).

4. In the current manuscript, the authors repeatedly described all results in the discussion section, which is undesirable. The authors need to rewrite the section of discussion.

5. Authors should examine the activation of CAR/RXR in vivo.

6. The manuscript needed to be checked by a native speaker. In particular, the abbreviations utilization is not standard.

7. The X-axis units of figure 1 confuse me. Is -9 M or 10-9 M?

Author Response

We thank the reviewer for his suggestions which improved the quality of the manuscript.

  1. The results of study revealed that the pesticides induced simple steatosis but did not significantly affect the dysfunction of the liver. In addition, the impacts of pesticides on the inflammation of liver are not known. I think 3 days is too short a time for pesticides treatment. Therefore, I suggest that prolonging the duration of drug treatment has yielded a more precise conclusion. I would also like to know the effect of pesticides on the body weight of mice.
  2. The authors should explain why these pesticides doses selected. How does the doses of these pesticides used in the current study compare with the residual in the environment?

#authors response to 1 and 2:  We thank the reviewer for pointing out these matters. This study was too short to yield good data for a precise conclusion on toxic risks during a realistic environmental exposure. The aim of this study was to be the first step for assessing possible combined effects of pesticides on CAR/RXR heterodimer and metabolic disruptions. To validate this hypothesis, a short acute exposure using few animals was sufficient. No change in body weight was observed and mentioned in the manuscript. Indeed, the study was too short to observe a potential effect on body weight. The chosen doses where higher than the daily admissible intake or environmental levels. But in the first instance, to have a better chance of observing effects, we chose to use higher doses to validate the concept. The doses were established following a dose response study for each pesticide and the highest dose which didn’t activate CAR was selected. Following the encouraging results of this experiment we launched a new experiment to study a low dose chronic exposure, with a pesticide mixture included in the food, more representative of human exposure allowing to precisely evaluate toxicological risks such as liver inflammation, fibrosis, lysis and body weight alterations.

  1. With the exception of Figure 3, the presentation of the results is confusing. I suggest that the tables and bar graphs also need to display as Figure 3 (i.e. clearly indicated the negative and/or positive control group).

#authors: Indeed this presentation can be a little confusing at first as the results are presented in fold change of the control group, therefore the control group always equal to 1 is not shown. To lighten the figures, already containing a lot of information, we decided not to include the negative control group which enables the reader to focus on the essential information.

  1. In the current manuscript, the authors repeatedly described all results in the discussion section, which is undesirable. The authors need to rewrite the section of discussion.

#authors: The discussion part has been rewritten with less description of results, more discussion items and enriched with more references.

  1. Authors should examine the activation of CAR/RXR in vivo.

#authors: We are very sorry but, we don’t understand this comment as this is precisely what we did. Maybe the reviewer meant “expression of CAR”? In that case we could say that CAR does not autoinduce its own transcription when activated. Therefore, the transcription level of CAR and it’s proteic levels observed in western blotting would not be the most relevant when studying the activation of this nuclear receptor. Western blot would give us information on CAR induction but not on car activation. Therefore, we used the CAR activation marker Cyp2b10.

  1. The manuscript needed to be checked by a native speaker. In particular, the abbreviations utilization is not standard.

#authors: English editing was performed

  1. The X-axis units of figure 1 confuse me. Is -9 M or 10-9 M?

#authors: -9 M correspond to 10-9. This has been corrected in Figure 1 and changed to 10-9 format.